# Geographical Variance in the Use of Tranexamic Acid for Major Trauma Patients

**DOI:** 10.3390/medicina55090561

**Published:** 2019-09-02

**Authors:** Kieran Walsh, Francis O’Keeffe, Biswadev Mitra

**Affiliations:** 1National Trauma Research Institute, The Alfred Hospital, Melbourne 3004, Australia; 2Critical Care Research, School of Public Health and Preventive Medicine, Monash University, Melbourne 3004, Australia; 3Emergency & Trauma Centre, Alfred Health, Melbourne 3004, Australia; 4Emergency Department, Mater Misericordiae University Hospital, Dublin D7, Ireland

**Keywords:** tranexamic acid, CRASH-2, trauma, major trauma, haemorrhage

## Abstract

*Background and Objectives*: The CRASH-2 trial is the largest randomised control trial examining tranexamic acid (TXA) for injured patients. Since its publication, debate has arisen around whether results could be applied to mature trauma systems in developed nations, with global opinion divided. The aim of this study was to determine if, among trauma patients in or at significant risk of major haemorrhages, there is an association of geographic region with the proportion of patients that received tranexamic acid. *Materials and Methods*: We conducted a systematic review of the literature. Potentially eligible papers were first screened via title and abstract screening. A full copy of the remaining papers was then obtained and screened for final inclusion. The Newcastle–Ottawa Scale for non-randomised control trials was used for quality assessment of the final studies included. A meta-analysis was conducted using a random-effects model, reporting variation in use sub-grouped by geographical location. *Results*: There were 727 papers identified through database searching and 23 manuscripts met the criteria for final inclusion in this review. There was a statistically significant variation in the use of TXA for included patients. Europe and Oceania had higher usage rates of TXA compared to other continents. Use of TXA in Asia and Africa was significantly less than other continents and varied use was observed in North America. *Conclusions*: A large geographical variance in the use of TXA for trauma patients in or at significant risk of major haemorrhage currently exists. The populations in Asia and Africa, where the results of CRASH-2 could be most readily generalised to, reported low rates of use. The reason why remains unclear and further research is required to standardise the use of TXA for trauma resuscitation.

## 1. Introduction

Trauma is the leading cause of death in developed nations for those under the age of 45 years [1,2]. Among injured patients reaching hospital, haemorrhages are the second leading cause of death—after central nervous system (CNS) injury—and the number one preventable cause of death [3,4,5]. Haemorrhages are complicated in 25–30% of severely injured trauma patients by trauma induced coagulopathy (TIC). TIC is characterised by hypocoagulability and hyperfibrinolysis and carries a poor prognosis and four-fold mortality rate compared to trauma patients without TIC [6]. Tranexamic acid (TXA) is the most widely used antifibrinolytic and it has been hypothesised that its routine use in trauma patients may result in reduced mortality by the correction of hyperfibrinolysis [7].

CRASH-2 is the largest randomised control trial (RCT) examining circulatory resuscitation for trauma patients to date. It enrolled 20,211 adult trauma patients from 274 hospitals in 40 different countries and randomised them into two groups—TXA (10,096) and placebo (10,115). The study found a statistically significant reduction in all-cause mortality in the TXA group (14.5%) compared to the placebo group (16%), with a relative risk (RR) of 0.91 (95% CI 0.85–0.97 *p* = 0.0035) [8]. CRASH-2 concluded that TXA administration within three hours of injury is safe, cost-effective and may result in a mortality benefit to injured patients [8].

However, the results of CRASH-2 have not been unanimously implemented into practice. Questions have been raised regarding its applicability to developed trauma systems that manage patients with high volumes of plasma and other blood products [9]. Only around half of the participants in CRASH-2 required a transfusion or operation, calling into question subject selection for the study and whether subjects in the study were truly at risk of a significant haemorrhage. A potential complication of TXA use within the trauma setting is thromboembolism [10,11]. Questions have been raised as to whether CRASH-2 adequately assessed for this complication. While the number of clinical events was recorded, no data on the number of patients who were tested for the condition was reported. It has been hypothesized that this is because they were not actively sought after and, thus, may be an underestimation of the true incidence [12]. Further questions regarding limitations of the CRASH-2 trial have since been raised including its approach to randomisation, the comparability of cohorts in regard to both Injury Severity Score (ISS) and shock and mortality rate in patients given TXA after 3 h [13,14].

To assess the translation of such a landmark RCT with positive results into clinical practice, we undertook a systematic review to assess the uptake of TXA into clinical practice after publication of the CRASH-2 trial. Specifically, we aimed to determine the association of regional variation in the use of TXA in trauma patients in or at significant risk of major haemorrhage.

## 2. Materials and Methods

All studies describing trauma patients in or at significant risk of major haemorrhage that reported TXA administration since the CRASH-2 trial and described resuscitation with blood and/or blood products and TXA were included. Included studies also had to report the primary outcome of interest, being mortality, at a defined time-point.

Excluded were animal studies, studies that included patients from before the CRASH-2 trial, studies that were not published in English and literature and systematic reviews that did not include original data about the proportion of trauma patients administered TXA.

A senior librarian was consulted to help develop a search strategy. OvidMedline and OvidEmbase were searched using a variety of subject headings and text words to identify relevant papers. Further texts were then identified using a combination of text word searching in Google Scholar and snowballing from reference lists of potentially eligible trials. The final search strategy is included in Appendix A and Appendix B.

Once identified, papers were assessed for final inclusion via a multi-tiered approach. Potentially eligible papers were first screened via title and abstract screening. A full copy of the remaining papers was then obtained and read through. Based on full text screening, a final list of papers for potential inclusion was discussed between researchers. Once deemed eligible for final inclusion, the quality of the remaining papers was analysed using the Newcastle–Ottawa Scale (NOS) and reported using their star-based system (Table 1). NOS uses a star-based system to assess the quality of non-randomised control trial studies. NOS assesses three criteria: selection (out of 4 stars), comparability (out of 2 stars) and outcome (out of 3 stars).

Data were initially extracted, and the results were formatted into a table using Microsoft word for Mac version 16.16.8 by a primary reviewer. The results of this table were then discussed among two co-authors. Post-discussion, additional data was searched for and papers were again reviewed by the same primary reviewer. Following this, data was extracted using Covidence Systematic Review Software. Information from the first table and the Covidence Systematic Review Software was then compared and results combined into the two final tables that have been included in this study. 

## 3. Results

There were 727 papers identified through database searching (Medline, *n =* 386; and Embase, *n =* 341) and 23 manuscripts met the criteria for final inclusion in this review. The PRISMA study flow diagram can be seen in Figure 1.

Out of the 182 papers where the full texts were read, 159 were excluded. 130 of these papers did not meet study design criteria for inclusion in this paper. A further 15 studies were excluded as they focused on a different study population—trauma patients who were not in or at significant risk of a major haemorrhage, or patients who were in or at significant risk of a major haemorrhage but for reasons other than trauma. Nine studies excluded data about patients who were eligible but did not receive TXA. As a result, calculating the proportion of eligible trauma patients who received TXA was impossible, therefore excluding them from this review. Finally, five studies were excluded as they included patients from before the publication of CRASH-2.

The final 23 studies included in the review all reported the proportion of trauma patients in or at significant risk of major haemorrhage who received TXA. A descriptive list of the 23 papers is presented in Table 2. All 23 are retrospective observational studies. Studies based in five different continents were included: Africa (*n =* 1), Asia (*n =* 4), Europe (*n =* 9), North America (*n =* 8), and Oceania (*n =* 1). Studies were included from both pre-hospital and emergency departments. Of the final 23 studies, 15 included information on cohorts’ average ISS. The majority of these studies had an average ISS of >12, indicating patients had experienced major trauma [15]. Both civilian- and military-based studies were included. The average age of military cohorts was younger in comparison to that of civilian studies, although average ISS was largely comparable. Only five studies reported their facility’s definition of ‘massive transfusion’ and the number of patients who received massive transfusion (MT). However, among studies that included a definition, the definition used was homogenous, with all five studies defining MT as a transfusion of ≥10 units of pRBCs.

Outcomes reported in selected manuscripts are presented in Table 3. The number of patients included in studies showed significant variance ranging from *n* < 10 to *n =* 7269. The number of patients eligible who went on to receive TXA also showed significant variance. Uptake was shown to be most predominant in Europe, with studies reporting up to 69% (95% CI: 68–70) of eligible patients receiving TXA [16]. 

Although included in the final tables, we excluded case studies and case series from meta-analysis and forest plot. We defined case studies and case series as *n* < 10 and therefore two studies were excluded. These were Aedo-Martin (2016) and Chesters (2015) [17,18].

The NOS chart results are reported in Table 1 below.

Figure 2 demonstrates the variance in proportion of eligible trauma patients that received TXA. We use authors’ self-reported definitions of what constitutes TXA eligibility. There was significant statistical heterogeneity. TXA use was reported to be most predominant in Europe and one study from Oceania. Use in the USA is currently mixed, and Asia and Africa report a low proportion of patients who receive TXA. Overall the reported proportion of TXA use among injured patients at significant risk or suffering major haemorrhage was 41% (95% CI: 25–57).

## 4. Discussion

This systematic review and meta-analysis of studies reporting on the use of TXA since the publication of CRASH-2 demonstrated substantial uptake of TXA use during trauma resuscitation. However, there is significant global geographical variance in the uptake of TXA for trauma patients. Awareness and understanding of the reasons for this variance are essential for appropriate translations of research findings.

One factor may be the perceived inadequacy of the level of evidence to support the routine use of TXA in the trauma setting. While CRASH-2 is a large multination RCT, and hence level 1 evidence, in a paper published after CRASH-2, 452 trauma surgeons across the USA completed an online survey addressing their use of TXA in the trauma setting. Of the 452, only 38.0% reported that they use TXA routinely in their practice [19]. Of the 72.0% who said they do not routinely use TXA, 47.7% reported that the reason for not using it regularly was ‘uncertain clinical benefit’. Furthermore, 10.0% indicated they believe there are better alternatives and 6.1% believe the risks outweigh the perceived benefits [19]. This supports the hypothesis that many physicians in mature trauma systems believe the results of CRASH-2 cannot be extrapolated to their clinical setting.

Knowledge gaps such as those identified in Napolitano, Cohen, Cotton, Schreiber and Moore [13], in their 2013 paper, are examples of the ‘uncertain clinical benefit’ to which these US trauma surgeons may have been referring.

Further debate exists around the implication of the results of the CRASH-2 trial. CRASH-2 has led some to the recommendation that TXA should be administered to all trauma patients within 3 h of injury [20]. However, this view is not universal, with even some of the authors of CRASH-2 suggesting that not all trauma patients may benefit from TXA [21].

A large observational study by Valle et al. [22] found increased mortality in trauma patients who received TXA. In their study, they compared 150 TXA patients to propensity-matched equivalents who did not receive TXA. The difference in mortality rates between the two cohorts failed to reach statistical significance. However, in a subgroup of patients who required emergency surgery within 30 min of arrival to hospital, they found a statistically significant two-fold mortality rate in patients receiving TXA compared to their non-TXA equivalents [22]. While this group is not reflective of the average trauma population, their results do add to the argument that TXA may not be universally beneficial for all trauma patients with major haemorrhages. 

Another factor contributing to the variance seen may be difficulty in translating the evidence into clinical practice. It is estimated that it takes an average of 17 years for evidence-based research to reach clinical practice [23]. While this is a rough and generalised average, it is reasonable to consider that the evidence from the CRASH-2 trial, published nine years ago in 2010, is in the ‘lag’ period between publication and clinical implementation. In their study Coats, Fragoso-Iniguez and Roberts [16] reported that TXA use increased every year, from 0% in 2010 to nearly 80% in 2016 [16]. This suggests that the results of CRASH-2 have not yet been fully realised and are still being translated into clinical practice.

This systematic review and meta-analysis have shown that the implementation of TXA since CRASH-2 has been most substantial in the UK. Following publication, UK-based participants of the CRASH-2 trial extensively promoted the trial using social-media, multi-media and medical education websites [24]. In 2011, a short Claymation video called “TRANMAN” was published by the London School of Hygiene and Tropical Medicine, promoting the use of TXA and encouraging people to review the results of CRASH-2 [25]. This was followed up by a song performed by a community choir, in which the lyrics encourage doctors to “save my life, use tranexamic acid” in response to a variety of hypothesised injuries [26]. Finally, a comic strip was commissioned and distributed to all EDs in the UK, in which emergency doctors use TXA for a variety of trauma scenarios, exclaiming its benefits [27]. The increased speed of translation of CRASH-2 into clinical practice in the UK, compared to other countries, may be due to the extensive promotion by some of the UK-based participants.

It must also be noted that the UK implements a method of additional funding via a “Best practice tariff” (BPT). A level 1 BPT of £1406 is made to hospitals for trauma patients with an ISS >8 and a level 2 BPT of £2819 for patients with an ISS >16, given that they fulfill 6 criteria, one of which is that TXA must be administered within 3 h of injury for patients requiring blood products [28,29]. It is reasonable to hypothesise that incentivisation of TXA administration with additional funding via a BPT may be one of the reasons for increased TXA administration within the UK.

The final factor we believe may be contributing to geographical variance is the ability of trauma systems to implement TXA into their clinical practice. While the WHO has added TXA to its list of essential medicines and it is widely available around the world, patients must present to a facility able to transfuse it within three hours of injury [8,30]. Despite being conducted in a developed nation with a mature trauma system, Bardes et al. [31] reported that 30.4% of all patients with an indication for TXA arrived to ED outside the 3 h window, deeming them ineligible for treatment [31]. Injured patients in rural settings and in nations with less mature trauma systems take longer to reach tertiary care facilities equipped to deal with their injuries [31,32]. This may limit the system’s ability to implement TXA, despite obvious clinical indications. Multiple nations have begun to implement pre-hospital TXA administration to overcome this barrier, however this practice is not universal and requires an advanced pre-hospital trauma system.

This is the first systematic review examining geographical variance in the use of TXA for trauma patients. CRASH-2 was published nine years ago and uptake appears to still be increasing. Therefore, it is plausible this data is an underestimation of current TXA use. It is also plausible that this systematic review suffers from publication bias. While grey literature was searched, only one unpublished paper was included in the final review. Published papers are more likely to show extreme results and may not be a true reflection of current practice. Furthermore, results of publications from single sites cannot be extrapolated and assumed to represent the wider geographical region. We know there is significant variation in the use of TXA based upon geographical location, it may also be possible that there is significant variation within these regions. Thus, results from published studies may only reflect use at those individual sites, rather than the region as a whole. We also acknowledge that the number of manuscripts published may not necessarily reflect drug use within a country.

This systematic review cannot postulate underlying reasons why significant geographical variance exists. Future research is needed to determine the cause of the variance seen in clinical practice. This systematic review also cannot determine the clinical benefit or harm of TXA use for trauma patients. Future high-quality evidence is needed to answer this question.

Two multinational RCTs are currently underway examining the use of TXA for trauma patients. These are the Pre-hospital Anti-fibrinolytics for Traumatic Coagulopathy and Haemorrhage Study (PATCH) and Study of Tranexamic Acid During Air Medical Pre-hospital Transport (STAAMP) trials. The Pre-hospital Anti-fibrinolytics for Traumatic Coagulopathy and Haemorrhage Study (PATCH) is an international, multicenter, double-blinded, placebo-controlled trial examining TXA in advanced trauma systems based out of Australia and New Zealand. PATCH will enroll 1200 severely injured patients deemed to be at risk of acute traumatic coagulopathy (ATC) using the coagulopathy of severe trauma (COAST) scoring system. The primary outcome of the PATCH trial will be the proportion of patients with a favourable outcome at 6 months, as defined by an extended Glasgow outcome score (GOSE) of 5–8 [33].

Study of Tranexamic Acid During Air Medical Pre-hospital Transport (STAAMP) is a US-based multicenter, double-blinded, placebo-controlled trial examining TXA in trauma patients. STAAMP will enroll 1000 severely injured patients at risk of a significant haemorrhage as defined by a pre-hospital systolic blood pressure (SBP) <90 or HR >110. The primary outcome is all-cause mortality at 30 days [34]. 

Both trials are currently in the recruiting stage but, upon completion, will potentially provide high-quality evidence regarding the benefits and harms of TXA implementation in a modern trauma system. It is hoped that this evidence will provide definitive answers to the debate that currently exists.

## 5. Conclusions

A large geographical variance in the use of TXA for trauma patients in or at significant risk of a major haemorrhage currently exists. However, the reason why remains unclear. Further studies are needed to explain the cause of this variance.

## Figures and Tables

**Figure 1 medicina-55-00561-f001:**
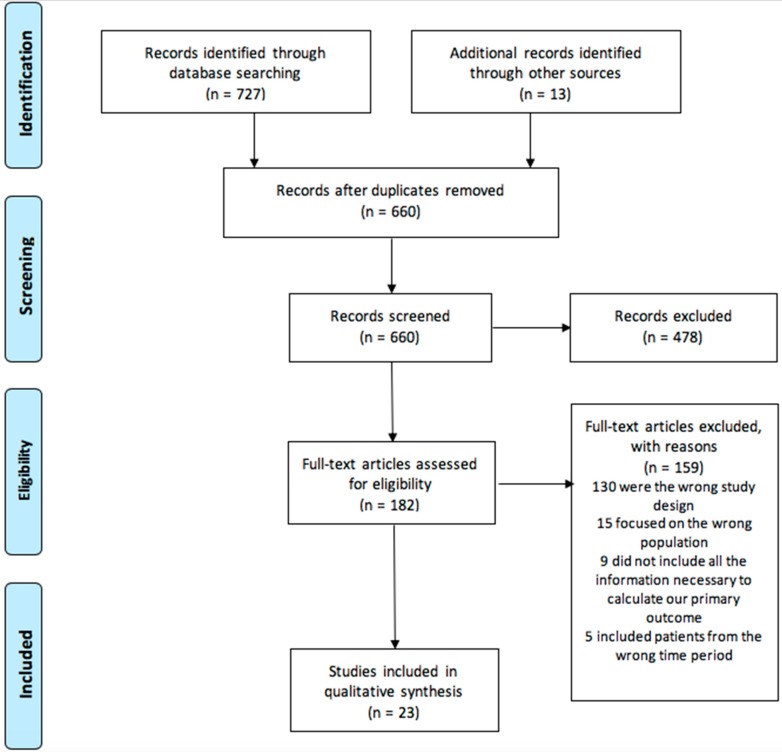
Prisma Flow Diagram.

**Figure 2 medicina-55-00561-f002:**
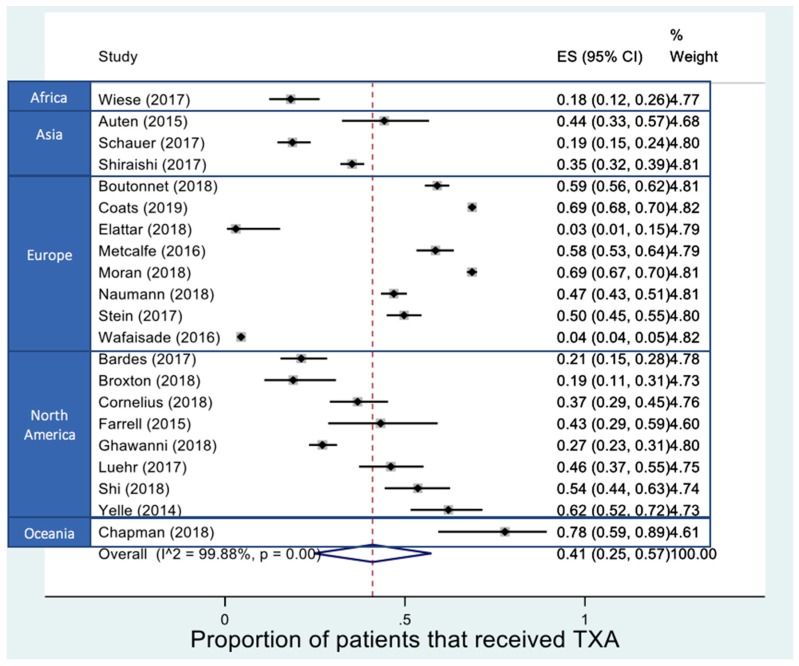
Forest plot of geographical variance of tranexamic acid (TXA) use for trauma patients in or at significant risk of major haemorrhage. (ES – Effect size, CI – Confidence interval).

**Table 1 medicina-55-00561-t001:** Newcastle–Ottawa Scale Quality Assessment of Manuscripts Included.

Author (Year)	Selection				Comparability	Outcome			Total
	Representativeness of Exposed Cohort (*)	Selection of Non-Exposed Cohort (*)	Ascertainment of Exposure (*)	Outcome of Interest Was Not Present at the Beginning (**)	(**)	Assessment of Outcome (*)	Was Follow up Long Enough for Outcomes to Occur (*)	Adequacy of Follow up (*)	(9 *)
Aedo-Martin (2016)	Explosive injuries not widely seen in another settings	*	*	*	**	*	*	*	8 *
Auten (2015)	*	*	*	*	**	*	*	*	9 *
Bardes (2017)	*	*	*	*	**	*	*	*	9 *
Boutonnet (2018)	*	*	*	*	**	*	*	*	9 *
Broxton (2018)	*	*	*	*	**	*	*	*	9 *
Chapman (2018)	*	*	*	*	*	*	*	*	8 *
Chesters (2015)	Undefined	*	Not Reported	*	* Patients may have gone onto receive tranexamic acid (TXA) in the Emergency department (ED)	*	*	Patients may have gone onto receive TXA in ED	6 *
Coats (2019)	*	*	*	*	**	*	*	*	9 *
Cornelius (2018)	*	*	*	*	**	*	*	*	9 *
Elattar (2018)	*	*	*	*	*	*	*	*	8 *
Farrell (2015)	*	*	*	*	**	*	*	*	9 *
Ghawanni (2018)	*	*	*	*	*	*	*	*	8 *
Luehr (2017)	*	*	*	*	**	*	*	*	9 *
Metcalfe (2016)	*	*	*	*	**	*	*	*	9 *
Moran (2018)	*	*	*	*	**	*	*	*	9 *
Naumann (2018)	*	*	*	*	* Patients may have gone on to receive TXA in ED	*	*	Patients may have gone on to receive TXA in ED	7 *
Schauer (2017)	*	*	*	*	**	*	*	*	9 *
Shi (2018)	*	*	*	*	**	*	*	*	9 *
Shiraishi (2017)	*	*	*	*	**	*	*	*	9 *
Stein (2017)	*	*	*	*	**	*	*	*	9 *
Wafaisade (2016)	*	*	*	*	* Patients may have gone on to receive TXA in ED	*	*	Patients may have gone on to receive TXA in ED	7 *
Wiese (2017)	*	*	*	*	**	*	*	*	9 *
Yelle (2014)	*	*	*	*	**	*	*	*	9 *

* indicates category is eligible for/has received one star; ** indicates category is eligible for/has received two stars.

**Table 2 medicina-55-00561-t002:** Study Design and Demographics.

*Author (Year)*	Setting—Level of Facility and Geographical Location	Setting—Pre-Hospital, ED, Post-ED	Mechanism of Injury	Inclusion Criteria of Trauma Patients Deemed Eligible for TXA	Median Injury Severity Score (ISS)	Mean Age (Years)	Definition of Massive Transfusion	How Many Patients Got a Massive Transfusion
Aedo-Martin (2016)	Spanish military hospital in Heart, Afghanistan	ED	*Blunt* - 0 *Penetrating* - 10	All casualties due to firearms or explosive devices that arrived at presented to ED	13	27.4	Undefined	Not reported
Auten (2015)	3 forward USA military surgical units in Helmand, Afghanistan	Pre-Hospital	*Blast* - 54 *Non-blast* - 7	USA military battle injuries with an ISS ≥16 who received transfusion of a blood product within the first 24 h of injury between January 2010 and July 2012	32 *	23.5	Undefined	Not reported
Bardes (2017)	Level 1 trauma centre in rural West Virginia, USA	Pre-Hospital, ED and Post-ED	*Blunt* - 152 *Penetrating* - 65	Patients for whom the trauma team had been activated with TXA indication, defined as any of the following: (1) Hypotension systolic blood pressure (SBP) <90 mmHg), (2) Receiving blood transfusions, (3) Initiation of the massive transfusion protocol, (4) Actively bleeding or (5) With a clinical concern for ongoing bleeding. If patients arrived >3 h post injury they were deemed ineligible for TXA, however were still included in calculating ‘mean age, ISS, MTP activation and blunt vs. penetrating trauma’.	24 *	45.5	Undefined	30
Boutonnet (2018)	Six Level 1 trauma centres in the Ile De France area, France	Pre-Hospital and ED	*Road accident* - 458 *Gunshot wound* - 31 *Stab wound* - 48 *Other* - 9	Adult (>16 years of age) trauma patients admitted directly to one of the six trauma centres between May 2011 and 31 December 2015, were eligible for the analysis. Patients were included if they had: (1) presented a major haemorrhage (defined by the need of four or more packed red blood cells [pRBC] in the first 6 h following the trauma), or (2) received at least one pRBC in the emergency room (ER), or (3) received vasopressors either in the prehospital setting or in the ER, (4) data required to calculate a propensity score available.	29	42	Undefined	Not reported
Broxton (2018)	Level 1 trauma centre in south eastern USA	ED and Post-ED	*Blunt* - 41 *Penetrating* - 17	Patients who received medical care for trauma injuries, patients who had any of the four indicators of massive bleeding documented in their Electronic Medical Record (EMR): (1) Administration of uncross matched PRBCs, (2) administration of TXA, (3) 4 or more units of PRBCs, given over 1 h, and/or (4) 10 or more units of PRBCs, given over 24 h; patients greater than 14 years of age.	Not Reported	44	Undefined	58
Chapman (2018)	Tertiary hospital in Christchurch, New Zealand	ED	Not reported	Trauma patients with an ISS >12 who required Massive transfusion protocol (MTP) activation in Christchurch Hospital ED from November 2015 to June 2017	21	45 †	Undefined	27
Chesters (2015)	Prehospital air ambulance in Lancaster, England	Pre-Hospital	*Blunt* - 6 *Penetrating* - 2	Patients classed as major trauma positive by the regional major trauma network triage tool with documented significant haemorrhage in the 2013 calendar year	Not reported	Not reported	Undefined	Not reported
Coats (2019)	Trauma patients across England and wales	Pre-Hospital, ED and Post-ED	Not reported	Trauma patients as defined by Trauma Audit and Research Network (TARN) criteria with an ISS ≥9 between 2010 and 2016 who received blood or a blood product transfusion within 6 h on injury	Not Reported	Not reported	Undefined	Not reported
Cornelius (2018)	Winged medical service and Level 1 trauma centre for southern USA: Louisiana, Texas, Arkansas and Mississippi	Pre-Hospital	*Blunt* - 120 *Penetrating* - 11 *Unknown* - 2	Patients who should have received TXA as per University Health–Shreveport’s guidelines between 2012–2016. That is trauma patients aged greater than 16 years, and 3 h or less since the time of injury with significant haemorrhage indicated by (1) hypotension (systolic blood pressure less than 90 mmHg) and/or (2) tachycardia (heart rate more than 110 beats per minute), and Injury Severity Score (ISS) 20 or greater	For TXA group 31. For non-TXA group 29. Overall not reported	37.9	Undefined	Not reported
Elattar (2018)	A teaching trust consisting of 4 hospitals in England, UK	ED and Post-ED	Not reported	Trauma patients presenting in a year (April 2015–March 2016) that met criteria indicating a high risk of bleeding as per CRASH-2 inclusion. Patients presenting post asphyxiation or drowning were excluded	Not reported	Not reported	Undefined	Not reported
Farrell (2015)	Level 1 and 2 trauma centres in Boston, USA	ED and Post-ED	*Blunt* - 10 *Penetrating* - 27	Trauma patients eligible to receive TXA during a 16-month study period (15 November 2012 to 15 March 2014), that is; (1) were trauma patients for whom the MTP had been initiated or should have been initiated, (2) arrived at BMC within 8 h of injury, (3) were 15 years of age or older, and (4) weighed at least 40 kg	13	36.2	Greater than 10 units of packed red blood cells within 24 h	34
Ghawanni (2018)	Level 1 trauma centre in Hamilton, Canada	ED	*Blunt* - 407 *Penetrating* - 77 *Other* - 11	Adult trauma patients (ISS ≥12 or an ISS <12 who the trauma team was activated for who (1) presented to the Hamilton General Hospital, over a two-year period between 1 January 2012 and 31 December 2014. (2) 16 years of age or older who (3) met one or more of the following criteria: (1) tachycardia (defined as a heart rate [HR] ≥110 beats per minute on arrival to the emergency department [ED]); (2) hypotension (defined as a systolic blood pressure [SBP ≤90 on ED arrival); and/or (3) requiring at least 1 unit of pRBCs in the ED	18 *	46.1	Undefined	Not reported
Luehr (2017)	Tertiary care facility in Springfield, USA	ED and Post-ED	*Blunt* - 96 *Penetrating* - 19	Trauma patients eligible to receive TXA between 2013–2016, that is: (i) >16 years old, (ii) no known hypersensitivity to TXA, (iii) no known severe renal failure, (iv) no known history of thromboembolism, (v) patient does not present with aneurysmal subarachnoid haemorrhage and (vi) patient is seen (and could have been administered TXA) by qualified medical personnel within 3 h of injury, (vii) All patients survived ≥8.5 h (minimum amount of time required to administer a full TXA dose), (viii) All patients received at least a single blood product.	20.9 *	41.8	Undefined	Not reported
Metcalfe (2016)	Newly developed regional trauma centres across England, UK	ED and Post-ED	Not reported	Trauma patients as defined by the TARN criteria in regional trauma centres in 9 months following their opening who were experiencing severe bleeding	Not reported	Not reported	Undefined	Not reported
Moran (2018)	35 hospitals with continuous TARN membership and patient submissions in England, UK	ED and Post-ED	Not reported	Trauma patients who met the TARN criteria (patients of any age who sustain injury resulting in: hospital admission >72 h, critical care admission, transfer to a tertiary/specialist centre or death within 30 days.) between 2010–2017 who also received transfusion with a blood product. Patients with an isolated femoral neck or single pubic ramus fracture >65 years and simple isolated injuries or an ISS < 9 were excluded	Not reported	Not reported	Undefined	Not reported
Naumann (2018)	Prehospital across 11 air ambulance organisations in England, UK	Pre-Hospital	*Blunt* - 654 *Penetrating* - 53 *Unknown* - 22	Patients were included if they had sustained a traumatic injury, were attended by a Pre-hospital emergency medicine (PHEM) team (which included a physician) and had a systolic blood pressure <90 mm Hg or absent radial pulse during their treatment and evacuation to hospital	Not reported	Not reported	Undefined	Not reported
Schauer (2017)	On the ground US prehospital medical treatment for combat casualties in Afghanistan during ‘Operation Enduring Freedom’	Pre-Hospital	*Explosive injury* - 121 *Gunshot Wound* - 138 *Other* - 15	Patients who were casualties in Afghanistan during Operation Enduring Freedom from January 2013 to September 2014 who should have received TXA as per tactical combat causality care (TCCC) guidelines (if a casualty is anticipated to need significant blood transfusion (presents with haemorrhagic shock, one or more major amputations, penetrating torso trauma, or evidence of severe bleeding)).	Only reported for patients from the Department of defense trauma registry (DoDTR) database (56) not PHTR database 20.1 *	Not reported	10 or more units of packed red blood cells in 24 h	Not reported
Shi (2018)	Level 1 trauma centre New England, USA	ED and Post-ED	Not reported	Trauma patients for whom the MTP was activated from 10 January 2014 to 31 October 2017.	27	40	≥10 units of packed red blood cells	92
Shiraishi (2017)	Multicentre study throughout Japan	ED and Post-ED	*Blunt* - 790 *Penetrating* - 6	Injured patients aged at least 18 years with an ISS of 16 or more who were admitted to one of the study hospitals	TXA group 25. No TXA group 22. Overall not reported	59	Undefined	Not reported
Stein (2017)	Level 1 trauma centre in Zurich, Switzerland	ED	*Blunt* - 394 *Penetrating* - 14	All adult trauma patients (≥16 years) with an injury severity score (ISS) ≥16, who were primarily admitted to the University Hospital, Zurich, Switzerland, between 2012 and 2014. Patients with missing records for initial emergency department treatment, with missing parameters that prevented calculation of the trauma associated severe haemorrhage (TASH) score [13,14], and/or secondarily transferred patients were excluded.	26	51	≥10 units of red blood cells from emergency department arrival until intensive care unit admission	15
Wafaisade (2016)	Pre-hospital emergency services and acute care across multiple German hospitals, Germany	Pre-Hospital	(Only TXA patients not whole population) *Blunt* - 90.3% *Penetrating* - 9.7%	Patients from databases Allgemeiner deutscher automobile-club (ADAC) and trauma register DGU (TR-DGU) collected between 1 January 2012 and 31 December 2014. Patients were included in this study according to the following criteria: 1. ADAC Air Rescue Service database: a. Primarily admitted trauma patient. Critical injury, defined as preclinically assessed National advisory committee for aeronautics (NACA) IV (potentially life-threatening), NACA V (acute danger) or NACA VI (respiratory and/or cardiac arrest). Admission to a trauma centre participating in the TR-DGU. 2. TR-DGU database: Primary admission and treatment in a German trauma centre (i.e., Exclusion of trauma centres from other countries).	For TXA group 24 *	For TXA group 43	≥10 units of packed red blood cells	For TXA group 10
Wiese (2017)	District public hospital in Cape Town, South Africa	ED	Not reported	Patients older than 13 years, who presented with an injury during twelve, randomly selected weeks eligible to receive TXA based on at least one of: (1) SBP <90 mmHg, (2) HR >110, and/or (3) Patients deemed to be at significant risk of haemorrhage based on ISS ≥16.	Not reported	Not reported	Undefined	Not reported
Yelle (2014)	Level 1 trauma centre Ottawa, Canada	ED	Not reported	Patients aged >16 and transfusion of at least 1u pRBC. With evidence of haemorrhage defined by at least one of; (1) multiple blood product transfusion (>1u pRBC or additional transfusion of another blood product), (2) SBP <90 mmHg, (3) HR >110 bpm, (4) Temperature <35.0 degrees Celsius, (5) Penetrating abdominal or chest wound, (6) Haemorrhage with GCS <13, (7) Amputation and/or (8) Severe extremity trauma	29	45.3	Undefined	Not reported

* Reported as mean instead of median; † Reported as median instead of mean.

**Table 3 medicina-55-00561-t003:** Study Results.

Author (Year)	Continent	Total Number of Patients Suitable for TXA	Number That Received TXA	Number That Did Not Receive TXA
Aedo-Martin (2016)	Asia	10	10	0
Auten (2015)	Asia	61	27	34
Bardes (2017)	North America	151	32	119
Boutonnet (2018)	Europe	797	470	327
Broxton (2018)	North America	58	11	47
Chapman (2018)	Oceania	27	21	6
Chesters (2015)	Europe	8	8	0
Coats (2019)	Europe	7269	4992	2277
Cornelius (2018)	North America	133	49	84
Elattar (2018)	Europe	33	1	32
Farrell (2015)	North America	37	16	21
Ghawanni (2018)	North America	495	134	361
Luehr (2017)	North America	115	53	62
Metcalfe (2016)	Europe	342	200	142
Moran (2018)	Europe	4238	2,909	1329
Naumann (2018)	Europe	729	342	287
Schauer (2017)	Asia	272	51	221
Shi (2018)	North America	112	60	52
Shiraishi (2017)	Asia	796	281	515
Stein (2017)	Europe	408	203	205
Wafaisade (2016)	Europe	5765	258	5507
Wiese (2017)	Africa	115	21	94
Yelle (2014)	North America	87	54	33

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
