# Peer review of "Geographical Variance in the Use of Tranexamic Acid for Major Trauma Patients"

_medicina, 2019, doi:10.3390/medicina55090561_

Round 1

Reviewer 1 Report

This paper aimed to look at the regional variation in the use of tranexamic acid (TXA). Unfortunately the authors used an odd method of doing this- by systematically reviewing the papers on TXA in the literature. They made the assumption that the number of papers and the practise in the published papers reflected regional practice. Moreover the number of papers publshed from different regions of the world is highly variable, fewere from Asia and Africa so of ocurse it would appear there were low rates of use of TXA on that basis

Secondly they have stated p2 lines 51/52 about CRASH-2 "A feared complication of TXA use, vneous thromboembolism was not adequately assessed with no data on the number of patients who were tested for the condition."

This is incorrect.

1) CRASH-2 published al lthe data on the incidence of clinical VTE in those with TXA and placebo and there was no difference

2) Patients were not routinely assessed for the presence of VTE in CRASH-2 but the number of clinical events is recorded. This is an accepted way of recording VTE

3) Thrombosis is NOT a feared complication of the use of TXA. Nearly a million patients have entered into prospective lcinical trials of TXA and none have reported increased rates of thrombosis. The only papers that do are retrospective studies where patients were not randomised to TXA. As the sickest patients tend to get TXA, the sickest group are also prone to get VTE.....

Reviewer 2 Report

The authors conducted a systematic review of the literature. The authors relied upon the “Newcastle-Ottawa Scale for non-randomised control trials which was used for quality assessment of the final studies included. They conducted a meta-analysis using a random-effects model, reporting variation in use sub-grouped by geographical location.” They undertook this study because a large geographical variance in the use of TXA for trauma patients in or at significant risk of major haemorrhage currently exists.

The authors found that the “population in Asia and Africa, where the results of CRASH-2 could be most readily generalised to, reported low rates of use. The reason why remains unclear and they recommended that further research is required to standardise the use of TXA for trauma resuscitation.”

I recommend minor revisions.

However, I have a few comments that the authors might consider incorporating into their manuscript.

There is a common comment which is repeated in the opening paragraph about the percentage of trauma patients who have TIC.  I recommend that the authors clarify this frequently mentioned percentage. Page 1 “Haemorrhage is complicated in 25-30% of patients by trauma induced coagulopathy (TIC).” It would be more appropriate to simply state “severely injured trauma patients”. The 25-30% figure is often quoted but the figure relates only to severely injured patients with high ISS scores.

In their discussion, the authors have referred to the “knowledge gaps” of the CRASH2 trial. They have not referred extensively enough to Napolitano’s et al excellent review that pointed very significant knowledge gaps that have not been addressed in the literature by the authors of the CRASH2 trial. Napolitano’s comprehensive review was not mentioned in the manuscript and this is a major omission that needs to be corrected. It should be included since the many points made by Napolitano et al were devastating in their criticism and these points of weakness of the CRASH2 trial have gone unanswered by the authors of the CRASH2 trial.

The articles of concern that I would include in the end notes as examples of the reasons that trauma surgeons view the CRASH 2 trial as “inadequate” are the following.

The US DOD Hemorrhage and Resuscitation Research and Development Steering Committee, “Tranexamic acid and trauma: current status and knowledge gaps with recommended research priorities,” Shock, vol. 39, no. 2, pp. 121–126, 2013.

L.M. Napolitano, M. J. Cohen, B. A. Cotton, M. A. Schreiber, and E. E. Moore, “Tranexamic acid in trauma: how should we use it?” Journal of Trauma and Acute Care Surgery, vol. 74, no.6, pp. 1575–1586, 2013.

The most interesting aspect of this failure of the rest of the world beyond England to adapt the recommendations of the CRASH 2 trial can be found in the second guessing of the CRASH 2 Trial results by the initial supporters and authors of the CRASH trial during the period of debate concerning the validity of the CRASH2 trial. I quote for the sake of accuracy from the paper by Binz et al. (reference 17)

 Cole et al., with CRASH-2 authors Davenport and Brohi, have recently acknowledged the “evidence gap” concerning the use of TXA in trauma in a study published in February 2015 in Annals of Surgery [20]. They recognize the limitation of the CRASH-2 trial stating that “the uptake of TXA use in civilian trauma has been variable, in part due to the difficulty in translating the results of these studies to mature trauma systems, with differences in study populations, logistics, and resource availability” [20]. This criticism has been raised many times prior to this acknowledgement [16, 17, 22]. Like Valle et al.’s retrospective, observational,

single-center study, which may be influenced by selection and surveillance bias, Cole et al.’s prospective cohort study is subject to a similar bias of small numbers and some insignificant outcomes, which may be different with a larger number of participants [19, 20]. Cole et al. analyzed 385 trauma patients admitted with an ISS >15, and patients were separated into shock and nonshock groups based on a base deficit ≥6mEq/L [20]. They were further divided based on TXA administration [20]. Among non-shocked patients, there was no difference in mortality among the TXA and no TXA groups [20]. Among shocked patients, the authors present different mortality rates based on the unadjusted data versus that arrived at by univariate analysis [20]. Based on the data without statistical manipulation, the authors conclude, “Unadjusted mortality rates between the 2 groups (shocked patients who received TXA and those who did not) were the same.” Concerning the shocked cohort, they add that “[e]arly mortality rates for those who had TXA were lower.  However, it should be emphasized that the unadjusted data does not show a statistically significant difference in either early or late mortality in the severely shocked patients even though the absolute number of deaths was less in the shocked

patients who received TXA[20].A result from the unadjusted data that was statistically significant was “a fourfold increase in thromboembolic events in the TXA group” among the shocked patients (TXA 8% versus no TXA 2%, ? < 0.01), confirming the concern for increased VTE with TXA use. The authors then performed a univariate analysis of the data, concluding that “TXA was independently associated with a reduction in MOF (multiple organ failure) and mortality in shocked patients and greater numbers of VFD (ventilator free days)” [20]. These statistical findings are presented because the subtle benefits require attention to the details of the analysis. What remains important, however, is that even with the use of subtle statistical justification of their conclusions,

Davenport and Brohi, in the Cole et al. study, as CRASH-2 authors, have shifted from their previous position of nearly universal use of TXA in trauma. They conclude the article by stating, “on the basis of the findings from the severely injured cohort in this study, it is difficult to recommend its use in nonshocked patients within mature civilian trauma systems” [20]. They ultimately recommend conditional TXA use, stating that “the findings give a clear signal for using TXA in severely injured, shocked civilian patients. ”However, careful analysis of their data suggests that TXA use in mature trauma systems does not reduce unadjusted mortality rates, further supporting the calls for more evidence on TXA use in trauma prior to its universal adoption in protocols.”

When even the coauthors of the CRASH2 trial publish results that contradict the findings of their own study, there is an understandable diminution of enthusiasm that initially met the much-celebrated release of the trial’s publication supported by a large campaign that initially encouraged universal utilization of TXA.

 The authors are to be congratulated for their integrity by demonstrating some of the methods that the CRASH 2 authors utilized in order to promulgate the results of this study filled with “knowledge gaps”.

However, there are many more irregularities that they have failed to mention in their study.

I again refer the editors and the authors to the full text of the Binz paper which unveils unsavory aspects of the CARSH2 propaganda enterprise

Specifically,” one of the authors of the trial, noted in an interview that “[s]cience is good at getting at the truth, but once you’ve found the truth, the methods that you use with science aren’t very good at getting the truth remembered. What we’re trying to do is get science to find the truth and use art to remember the truth” [32]. With this rationale, he began a campaign to increase the use of TXA in trauma by turning to “more innovative than traditional methods” [32]. ” These efforts spanned many multimedia milieus. The authors have referred to these milieus only briefly. Yet there were more than milieus involved. There was a level of propagandizing that is quite unusual in traditional medicine.

In addition, the many of the same authors as the CRASH2 trials have published the WOMAN trial which also suffered similar methodological deficiencies in design and implementation. Shakur H, Roberts I, Fawole B, et al. Effect of early tranexamic acid administration on mortality, hysterectomy, and other morbidities in women with post-partum haemorrhage (WOMAN): an international, randomised, double-blind, placebo-controlled trial. The Lancet. 2017;389(10084):2105-2116.

The WOMAN trial failed to be reproduced most recently in the large RCT by Sentihels et al who showed the lack of efficacy of TXA to reduce mortality when given prophylactically to post-partum women. To be specific, the conclusion of the WOMAN trial was the following: “Among women with vaginal delivery who received prophylactic oxytocin, the use of tranexamic acid did not result in a rate of postpartum hemorrhage of at least 500 ml that was significantly lower than the rate with placebo.” Sentilhes L, Winer N, Azria E, et al. Tranexamic acid for the prevention of blood loss after vaginal delivery. New England Journal of Medicine. 2018;379(8):731-742.

The WOMAN trial was by many of the same authors as the CRASH 2 trial with similar small benefit noted in austere environments where control of variables and adherence to strict RCT protocol may have been variable.

Finally, the uptake in England of the CRASH 2 trial results may have been significantly influenced not just by the “boosterism” and social media and television, radio and other forms of media written after the CRASH 2 trial, (See Binz et al reference 17) but also by the punitive penalty of failing to give TXA to trauma patients in England. Specifically, it has been noted that:  ” For example, NHS England have put a system in place so that tranexamic acid will be available in all road and air ambulances, to be given at the roadside if necessary; and under the “payment by results” system, unless tranexamic acid is given to all appropriate trauma patients, units will not be paid.” Hunt B, Raza I, Brohi K. The incidence and magnitude of fibrinolytic activation in trauma patients: a reply to a rebuttal. Journal of Thrombosis and Haemostasis. 2013;11(7):1437-1438.

In summary, the authors have submitted an important work that demonstrates the variable “uptake” of the results of the CRASH 2 Trial. They have demonstrated the geographic variability of the uptake. However, they could have bolstered their discussion with reference to the above-mentioned facts of:

1.       Initial campaign that not only encouraged use in trauma but subjected those in England who did not use TXA in trauma resuscitation to financial fines.

2.       The “specific knowledge gaps”, most specifically noted by Napolitano et al, that the authors have not mentioned.

3.       The retreat by the CRASH 2 authors themselves regarding the utility f TXA in mature trauma systems.

4.       The unusual and intense campaign in many venues and through many media milieus  following the CRASH2 trial release. The authors only scratched the surface of the very unusual and nonscientific methods of propaganda that were used to promulgate a study that had at best: “meagre demonstration of benefit. They have demonstrated just a few of the campaign irregularities following the release of the CRASH 2 trial.

5.       The subsequent failure of the WOMAN Trial to be validated in a large RCT follow up study

I suspect that the authors are aware of the `many irregularities following the release of the CRASH 2 trial that have been repeated for the WOMAN trial which have been justified under the banner of the Randomized Controlled Trial.  I would imagine that the authors wish to avoid the unpleasant repetition of complaints of many about the irregular propagandization of the CRASH2 results and now of the WOMAN trial results. However, failure to confront bias, not just in the scientific arena, but also in the post publication arena is not a salutary standard. I recommend that the authors delve more deeply into the universally recognized flaws of the CRASH 2 trial which would help explain the irregular uptake of the CRASH 2 trial results.

I suggest that the authors add the above-mentioned references and abbreviated discussions be included in their manuscript in order to lend some sense of cause to the failure of the CRASH2 trial to achieve greater uptake.

In the meantime, we must wait for the results of the TXA trials that the authors mention.

Interestingly, the most recent large observation trial from Pittsburg, the cite if the STAMP Trial, has demonstrated a significant increase in VTE among patient given TXA for trauma resuscitation. This study as well, bears mentioning. 

“Of 21,931 patients, 189 pairs were well matched across propensity score variables (standardized differences <0.2). Median Injury

Severity Score was 19 (interquartile range, 12–27) and 14 (interquartile range, 8–22) in TXA and non-TXA groups, respectively (p = 0.19). Tranexamic acid was associated with more than threefold increase in the odds of VTE (aOR, 3.3; 95% CI, 1.3–9.1; p = 0.02). Tranexamic acid was not significantly associated with survival (aOR, 0.86; 95% CI, 0.23–3.25; p = 0.83). Risk of

VTE remained elevated in the TXA cohort despite accounting for mortality (subdistribution hazard ratio, 2.42; 95% CI, 1.11–5.29; p = 0.03).

CONCLUSION: Tranexamic acid may be an independent risk factor for VTE. Future investigation is needed to identify which patients benefit most

from TXA, especially given the risks of this intervention to allow a more individualized treatment approach that maximizes benefits and mitigates potential harms. (J Trauma Acute Care Surg. 2019;86: 20–27. Copyright © 2018 American Association for the Surgery of Trauma)”

Myers SP, Kutcher ME, Rosengart MR, et al. Tranexamic acid administration is associated with an increased risk of posttraumatic venous thromboembolism. J Trauma Acute Care Surg. 2019;86(1):20-27.

The authors are to be congratulated for their objective demonstration of the variable uptake of the results of the CRASH2 trial. What is lacking is a demonstration of the reasons why trauma surgeons have not embraced the utilization of TXA in trauma resuscitation. This needs to be addressed in more detail in their manuscript. No doubt the authors wish to avid controversy. However, they have already touched on the subject with their reference to the unusual methods of proselytizing by the authors of the CRASH2 trials in the post publication period. There is much more to be described which can explain the reluctance by many trauma surgeons to accept the results of the CRASH 2 trial. I suggest that the authors enlarge on these “knowledge gaps” and gaps in scientific propriety in order to more clearly explain why trauma surgeons have not embraced the CRASH2 trial.

Thank you

Round 2

Reviewer 1 Report

The authors have not addressed my points at all!

1) They must point out that the number of papers published about a drug  by a country does not necessarily reflect use of the drug in that county

2) They do not address the issue that there is NOT an increased risk of thrombosis with the use of TXA but refer to scientifically inferior studies (small retrospective reviews and cohort studies ) that are superceded by the large RCTs using TXa. e.g. the WOMAN study.. Their arguement that in CRASH-2 there was no active searching for thrombosis is not a good argument. CRASH-2 looked at the rates of CLINICAL VTE at 30 days in both those treated with TXA and placebo.
